# Barriers and Facilitators to Accessing Mental and Physical Health Care Among Sexual Minority Women: A Qualitative Exploration

**DOI:** 10.3390/ijerph22060965

**Published:** 2025-06-19

**Authors:** Charlotte A. Dawson, Alicia Moulder, Kristin E. Heron

**Affiliations:** 1Macon and Joan Brock Virginia Health Sciences, Old Dominion University, Norfolk, VA 23507, USA; 2Virginia Consortium Program in Clinical Psychology, Norfolk, VA 23529, USA; amoul001@odu.edu (A.M.); kheron@odu.edu (K.E.H.); 3Department of Psychology, Old Dominion University, Norfolk, VA 23529, USA

**Keywords:** sexual minority women, barriers, health care, discrimination, mental health, health disparities

## Abstract

Cisgender sexual minority women (SMW, e.g., lesbian, queer) are at greater risk for poor mental and physical health compared to heterosexual women and face challenges when accessing health care. Previous research has largely focused on general sexual and gender minority barriers to health care, but more research is needed on the experiences of specific subgroups, including cisgender SMW. The current study qualitatively explored barriers and facilitators for cisgender SMW seeking health care. Twenty cisgender SMW aged 18–40 recruited using Meta advertisements and past participant lists completed 45 min semi-structured interviews and a brief survey. Thematic analysis conducted by two coders revealed a barrier theme with six subthemes, and a facilitator theme with seven subthemes. The barrier subthemes included discrimination, dominant culture centric, unsupportive socio-political environment, lack of patient-centered care, avoidance/concealment of sexual identity, and socio-economic challenges. The facilitator subthemes included supportive socio-political environment, advance identification of LGBTQ-affirming HCPs, patient-centered care, HCP identity similar to patient, social support, re-engagement with care after bad experiences, and socio-economic advantages. This study provides insight into the lived experiences of cisgender SMW that can help improve knowledge about health care disparities and inform health care interventions for this population.

## 1. Introduction

Healthy People 2030 [1], the National Academies [2], and The Institute of Medicine [3] have all highlighted the importance of continued research on the unique health concerns among sexual and gender minority (e.g., gay, transgender, etc.) populations. Sexual minority women (SMW, e.g., lesbian, queer, bisexual, etc.), an important sexual and gender minority subgroup, face a number of health disparities relative to their heterosexual peers, including elevated mortality risk [4] and suicidality [5]. Compared to their heterosexual counterparts, SMW are more likely to report poor mental health [6,7,8], obesity and overweight [9,10], and health conditions such as diabetes, high blood pressure, high cholesterol, and cancer [11]. Furthermore, bisexual women appear to be at particular risk for mental and physical health concerns, including sexual and reproductive health issues [12,13].

### 1.1. Health Care Disparities Among SMW

In addition to health outcome disparities between SMW and heterosexual women, there are also disparities between these groups when it comes to health care, such as unmet health care needs and negative experiences when receiving health care. Gay, lesbian, and bisexual women are more likely to report delays in receiving care or not receiving care due to cost and are also more likely to report not having a regular source of health care compared to their heterosexual counterparts [14]. Bisexual women, in particular, appear to experience unique health care concerns relative to gay, lesbian, and straight women. They are significantly more likely not to receive specific health services due to cost, to delay care for reasons unrelated to cost, and to experience difficulty finding a medical professional [14]. On top of facing more challenges when accessing health care, SMW may also encounter more difficulties when receiving care. Sexual minority individuals are 1.5 times more likely to endorse negative health care experiences compared to heterosexual individuals [15]. Compared to heterosexual women, lesbian and bisexual women report lower confidence in and poorer communication with health care professionals (HCPs) [15].

Sexual minority individuals experience discrimination, prejudice, and stigma related to their sexual identity that impact their health and health care experiences [16,17]. These minority stress processes are at the root of many barriers to care faced by SMW. A systematic review of lesbian, gay, bisexual, and transgender (LGBT) access to health care identified the following barriers: identity-related discrimination, presumed heterosexuality, humiliation and rejection, lack of LGBT-specific professional training, and internalized homophobia (i.e., applying negative views of others about non-heterosexual people to oneself) [18]. In summary, sexual minority discrimination hinders access to health care for SMW.

Due to disparities in health care for this population, previous research has also investigated ways to facilitate appropriate health care for SMW. Facilitators for sexual minority individuals include their own resilience (i.e., willingness to re-engage with services after discriminatory or otherwise negative experiences) and health-promoting engagement (e.g., committing to preventive screenings) [19]. HCPs can create affirmative spaces, develop trusting relationships, and ensure privacy to better care for sexual minority individuals [20]. Resilience and affirmative health care practices may help mitigate some of the barriers to health care for SMW.

#### 1.1.1. Mental Health Care Disparities Among SMW

An area of health care where SMW may face particular challenges in accessing equitable care is mental health care. LGBT individuals are more likely to have unmet mental health needs compared to heterosexual individuals [21]. Specifically, bisexual women are about two times more likely to report an unmet mental health care need compared to their heterosexual counterparts [22]. Additionally, mental health care programs that focus on LGBT care may be limited. The 2016 National Mental Health Service Survey reported that only 12.6% of state-approved mental health facilities had LGBT-specific programs [23].

#### 1.1.2. Sexual and Reproductive Health Care Disparities Among SMW

There are also significant differences in sexual and reproductive health care between SMW and heterosexual women. For example, SMW experience inappropriate reactions and rejections from gynecologists and report lower frequencies of cervical and breast cancer preventive screenings [18]. Regarding cervical cancer screening, Black LGB women report fears of discrimination and concerns related to heteronormative assumptions and lack of LGB knowledge among HCPs [24]. There are also differences between bisexual and lesbian women concerning sexual and reproductive health. Bisexual college women were more likely to undergo gynecological examinations, be tested for HIV, and self-examine their breasts compared to lesbian college women [25]. Sexual and reproductive health care disparities impact SMW, with unique challenges for both bisexual and lesbian women.

### 1.2. Using Models of Access to Health Care to Understand Access Disparities for SMW

Andersen and Davidson’s [26] health care access model can help guide exploration and understanding of disparities in health care. Their model was designed to examine various factors influencing health care access among all populations. However, it can also be applied to specific populations, such as SMW, to gain a deeper understanding of the unique barriers and facilitators they encounter in accessing care. In Andersen and Davidson’s [26] health care access model, contextual factors refer to the social, environmental, and policy-related influences that shape an individual’s ability to access health care. Also evaluated are individual factors, or personal characteristics, that influence access to care.

Contextual factors include social determinants of health, such as community safety or support [26]. For example, SMW who live in areas with strong LGBTQ+ community support may have better access to LGBTQ+ focused care, while those in less inclusive environments may not have access to specialized HCPs. Additionally, cultural and societal norms (i.e., societal attitudes toward health and health care) play a role in how health care is perceived and accessed. Cultural and societal norms around sexual identity also influence SMW’s health behaviors, with stigmatization leading to risk for delayed or avoided care. Health care system characteristics, such as the availability of HCPs and organizational support, also fall under contextual factors, as they directly affect the accessibility and quality of care. Last, public health policies and laws (e.g., protections against discrimination) can either facilitate or hinder access to health care by specific populations.

The model also emphasizes individual factors, personal characteristics that influence whether an individual seeks and utilizes health care services [26]. These factors include demographic characteristics (e.g., age, gender, and race/ethnicity), which can affect one’s health behavior and likelihood of seeking care. Health beliefs and attitudes also play a significant role in determining individual health care utilization. For example, some SMW may avoid health care settings due to previous negative experiences, reducing their likelihood of engaging in preventive care. Resources (i.e., income, health insurance) and support systems allow individuals to access care. Finally, an individual’s perception of their health or diagnosis influences their decision to seek care.

Taken together, the interaction between contextual and individual factors in Andersen and Davidson’s [26] health care access model is essential for understanding how SMW navigate health systems. Factors such as societal attitudes, laws and policies, personal experiences, and availability of resources all impact how SMW perceive their health needs and access care, in turn influencing their health outcomes.

### 1.3. Summary of Research Gaps

Overall, it is well documented that SMW are more likely to experience poor mental health [8], obesity [9], chronic conditions such as diabetes and high blood pressure [11], and higher mortality rates [4]. They also encounter numerous health care access challenges, including cost-related barriers, delays in care, and negative experiences with HCPs. Minority stressors (e.g., discrimination) exacerbate these challenges [16,17]. Additionally, SMW face unique mental health and sexual and reproductive health disparities, such as more unmet mental health needs and difficulties accessing preventive screenings. Andersen and Davidson’s [26] model of health care access can be used to examine the complex factors affecting SMW’s ability to access care, including social, cultural, and policy-related influences, as well as individual factors such as socio-economic status, health beliefs, and previous health care experiences.

Given the historical exclusion of SMW from research [27] and the unique stressors they face [16,17], it is essential to listen to SMW’s own accounts of their health care experiences. Lawson and Marsh [28] emphasized the value of qualitative research in improving the health of underserved women by amplifying their voices and supporting the development of patient-centered care. By exploring the lived experiences of underserved women, interventions that are more attuned to the specific needs of these populations can be created and implemented.

### 1.4. Current Study

The aim of the current study was to identify the barriers and facilitators that SMW experience when seeking and accessing mental and physical health care, using insights from in-depth interviews with a diverse group of SMW. By capturing their personal experiences and perspectives, this study aimed to provide barrier and facilitator themes that can inform health care practices, policies, and interventions tailored to the needs of SMW. It was hypothesized that barrier themes would include discrimination and heterosexism [18], and that facilitators would include resilience, trusting relationships, and affirmative spaces [19,20]. It was also predicted that participants would experience barriers in the areas of mental health [21,22,23] and sexual and reproductive health [24,25].

## 2. Methodology

### 2.1. Participants

SMW living across the United States were recruited through lists of past participants of previous research studies who had agreed to be contacted for future research, as well as through Meta advertisements. Participants were included if they (1) identified as sexual minority (lesbian, bisexual, gay, queer, pansexual, etc.); (2) were between the ages of 18 and 40; and (3) were a cisgender woman. Participants were excluded if they identified as heterosexual or straight. There were not any inclusion criteria specific to health behaviors, given the exploratory nature of this study. Twenty participants (*M*age = 28.50, *SD* = 4.26) meeting the inclusion criteria described participated.

Half of the participants (*n* = 10) identified as White (non Latina). The remaining participants identified as Black (non Latina; *n* = 4), Black (Latina; *n* = 1), White (Latina; *n* = 1), White and Asian (*n* = 1), Latina (*n* = 1), Asian (*n* = 1), and Asian and Hawaiian/Pacific Islander (*n* = 1). Regarding sexual identity, participants were able to select multiple identities, and they identified as Lesbian (*n* = 15), Queer (*n* = 11), Gay (*n* = 8), Pansexual (*n* = 5), and Bisexual (*n* = 4). The participants’ body mass index ranged from 19.60 to 68.70 (*M* = 34.55, *SD* = 13.34).

### 2.2. Procedures

This study was conducted in accordance with the Declaration of Helsinki and approved by the Institutional Review Board at Old Dominion University. Potential participants completed a brief online screening survey through Qualtrics. After the participants completed the screening survey, they were then informed via email if eligible for this study. Semi-structured interviews were conducted with the participants via a secure Zoom link by a trained research assistant from December 2021 to February 2022. At the beginning of the interview, the research assistant reviewed the study notification document with the participants, and the participants were given the opportunity to ask questions and decide whether they wanted to participate in the interview. The participants were added in increments of 5 until saturation was reached, based on recommendations that 20–30 interviews are typically sufficient to meet saturation [29]. The length of the interviews was approximately 45 min. The interviews were recorded via Zoom with participant consent to include visual and audio content. Zoom was used to generate interviews transcriptions, and a trained research assistant reviewed the transcriptions to ensure their accuracy. After the interview, the participants completed a brief demographics survey. Participants were given a USD 30 Amazon gift card for completing the interview and demographics survey.

### 2.3. Materials

An overview of interview topics and questions and a copy of the demographics survey can be found in Appendix A. The interview questions were developed using Andersen and Davidson’s [26] model, along with barriers and facilitators to health care access that have been identified in past literature. The demographics survey assessed participant age, race/ethnicity, height and weight, sexual identity, as well as other factors related to sexual orientation.

### 2.4. Data Analysis

Thematic analysis was used as a framework to analyze the interviews using both inductive and deductive strategies. Braun and Clarke’s guide to thematic analysis in psychology was utilized [30], including the six phases highlighted below. Two researchers transcribed the interviews and reviewed the interview transcripts, first by becoming more familiar with the data (Phase 1) and making notes of initial codes using NVivo 12 software (Phase 2). Then, two independent coders searched for broader themes (Phase 3) and subthemes and coded relevant quotes under the themes. Each theme, subtheme, and coded quote was reviewed, and collaborative decision making was used when there were differences (Phase 4). Next, the themes and subthemes were further defined and refined by clarifying theme names and relevant codes (Phase 5). Last, the analysis of the themes was written up in the Findings section below (Phase 6). The demographic survey data was analyzed using SPSS version 29.0.1, with descriptive statistics being conducted on these data to characterize the sample.

## 3. Findings

The two major themes identified by the researchers included barriers (Table 1) and facilitators (Table 2). Six barrier subthemes were identified, with four contextual subthemes (discrimination, dominant culture centric, unsupportive socio-political environment, and lack of patient-centered care) and two individual subthemes (avoidance/concealment of sexual identity and socio-economic challenges). Seven facilitator subthemes were identified, including four contextual subthemes (supportive socio-political environment, advance identification of LGBTQ-affirming HCPs, patient-centered care, and HCP identity similar to that of the patient) and three individual subthemes (social support, re-engagement with care after bad experiences, and socio-economic advantages).

### 3.1. Barriers

Based on the interviews of 20 SMW, six barrier subthemes were identified, including discrimination, dominant culture centric, unsupportive socio-political environment, lack of patient-centered care, avoidance/concealment of sexual identity, and socio-economic challenges.

#### 3.1.1. Discrimination

Participants experienced prejudice or discrimination from HCPs or support staff related to their sexual identity. Two participants shared times when they did not receive care because of their sexual identity. Participant 3 (White) stated, “I have called a [doctor’s] office just to make sure they were LGBT affirming and asked around a lot, and then they called me to cancel my appointment.” Participant 14 (White) shared discrimination related to religion and sexual identity, stating, “I encountered this one physician, where he was just … a strict Christian and … he just said like ‘I can’t treat patients who are against my religious belief.’” Participants also reported experiences of discrimination related to gender, weight, and race.

#### 3.1.2. Dominant Culture Centric

Heteronormativity, or the assumption or belief made by HCPs that heterosexuality is the norm was a code identified within the dominant culture-centric subtheme, in addition to lack of LGBTQ+ health knowledge among HCPs and participants’ families. One participant encountered heteronormativity regarding sexual behavior influencing her access of reproductive healthcare:

“You know you go to the doctor and they’re asking you … if you’re sexually active, ‘Are you on birth control? What’s your methods [of birth control]?’ They don’t stop to ask ‘Okay, who are your partners? What is your, you know, lifestyle like?’ And so it’s always an awkward conversation.”(Participant 1, White)

Participants identified HCP lack of LGBTQ+ specific knowledge as a challenging factor in accessing care. Participant 22 (Black/African American) described HCP difficulty adapting care to her as a SMW, “A lack of knowledge [from HCPs] about how something might specifically apply to … my identity as a queer woman, as opposed to just being straight … especially in terms of sexual health.” Participant 14 (White) highlighted general uncertainty from HCPs, “It definitely takes them [HCPs] some time to process, and I think some of them really try to accommodate you, but it’s definitely something that they’re like ‘Yeah now what do I do with you?’”

#### 3.1.3. Unsupportive Socio-Political Environment

Another barrier subtheme identified was living in an area that is not supportive of SMW. The codes within this subtheme included stigmatizing laws and policies and a perception that the town they live in is not accepting. Participant 2 (White) described how people in her area reacted negatively to a Pride parade:

“We had like a Pride parade … It was posted on a Facebook group … I’m in these Facebook groups so I can like learn what’s going on in the community for events … and the comments were just really mean, and it was kind of disheartening … because I really kind of thought, things were getting better, but people were really mean.”

#### 3.1.4. Lack of Patient-Centered Care

SMW identified HCPs not listening as an area of concern. Participant 4 (White) stated, “I mean it’s important to go to doctors, but also if I’m not going to get … actually listened to and treated and everything, then what’s the point of even going?” Participant 22 (Black/African American) shared an experience of perceiving that her questions about her health and care were not being answered:

“The only thing that I would classify as negative is … maybe have a doctor who wasn’t explaining to me … exactly what was going on in my body, even though I was asking. Or not fully explaining to me what the medication would do. I’m someone who likes to really fully understand things in detail, so maybe not having people answer my questions.”

#### 3.1.5. Avoidance/Concealment of Sexual Identity

Participants reported concealing their sexual identity from HCPs, with some avoiding treatment altogether to avoid disclosing their sexual identity. Other SMW noted avoiding treatment due to previous negative experiences in health care. Participant 17 (Black/African American) described her thought process behind avoidance/concealment of her sexual identity:

“I don’t want to be treated differently. So I just kind of hope that they don’t ask [about sexual identity] because … what is the point of you knowing that? especially as … a provider that’s not dealing with my sexual health. So I try to avoid that question, or like skirt around the answer and try to answer in a very general way.”

#### 3.1.6. Socio-Economic Challenges

The participants highlighted how low income and lack of health insurance impacted their ability to access health care. Participant 10 (White, Latina) described how her experience growing up in a low-income family influenced her experiences and perspectives on health care:

“I was kind of raised lower income. It’s just been generalized that … you don’t go to doctor until you’re dying … I’ve never gotten annual checkups … even after I broke my finger my parents made me wait a day to see if it sorted itself out before we actually went to the doctor.”

### 3.2. Facilitators

The SMW participants endorsed the following facilitators: supportive socio-political environment, ability to identity LGBTQ-affirming HCPs ahead of time, patient-centered care, sharing an identity with a HCP, social support, continuing to engage in health care in spite of negative experiences, and socio-economic benefits.

#### 3.2.1. Supportive Socio-Political Environment

Participants highlighted the importance of having a supportive environment, including having an inclusive health center close to them. These health centers included those that explicitly identified as LGBTQ+-affirming, Planned Parenthood, and university health and counseling centers. Participant 3 (White) stated, “So a major hospital system … just opened a new LGBT+ specific health care facility so that’s been … huge. That’s been monumental here.” Participant 11 (White) shared her experience with Planned Parenthood, saying, “Planned Parenthood, that’s always been very, very good for LGBTQ people to use. I’ve always used them as a gynecologist, and I’ve always felt incredibly comfortable.” Another indicator of supportive environments was strong LGBTQ+ presence/visibility in the community. Many SMW noted that seeing support for or members of the LGBTQ+ community in their area increased comfort. Participant 5 (White, Asian) shared how seeing inclusive advertisements impacted her:

“Yeah … definitely being on the subway and seeing all these advertisements for PrEP [pre-exposure prophylaxis] is huge. I’m seeing even just like vaccination PSAs [public service announcements] and having queer people in them. I guess just the visibility overall kind of makes it easier, so it’s not like, ‘Oh I’m gonna get in trouble for going to a doctor and asking for information on STIs that you know are more prevalent in the queer Community,’ things like that.”

Participant 11 (White) described her motivation for moving to an inclusive community:

“I really see a lot of queer couples out and about you know holding hands. There’s flags in the windows in the neighborhood, and I’m very happy because that’s definitely one of the reasons I moved because I wanted it to feel you know more comfortable, and I live with my girlfriend.”

#### 3.2.2. Advance Identification of LGBTQ-Affirming HCPs

Participants reported that knowing whether an HCP was affirming ahead of time helped in accessing health care. Some participants noted that researching HCPs online was helpful. Participant 1 (White) stated, “I found my therapist online and was able to read her bio and see that she was inclusive and an ally, and … it kind of took down that barrier to reach out and say hey.” Participant 9 (White) reported relying on recommendations from others in the LGBTQ+ community stating, “the ‘Keep [city name] Queer’ people will post in the [social media group] if they need help, or like a lot of times, people are seeking out a therapist or a licensed clinical social worker or psychiatrist or doctor etc.”

#### 3.2.3. Patient-Centered Care

Another facilitator subtheme identified by SMW was patient-centered care. Some participants recognized the significance of HCPs who listen and are supportive, with Participant 6 (White) stating the following:

“I’ll be like, ‘I have a wife’ and that’s not concerning [to the HCP] but then I can also bring up things like if I’ve got a sexually based concern they support my particular circumstance, as well as supporting just concerns in general.”

Participants also highlighted that when HCPs ask inclusive questions it can improve their health care experience. Participant 2 (White) noted, “I’ve been to some doctors that specifically asked me … women or men [regarding sexual behavior] and that I guess made me feel a little better because they acknowledge that it exists.”

#### 3.2.4. HCP Identity Similar to That of the Patient

The participants reported how having a HCP with a similar identity, including gender identity, sexual identity, and racial identity, impacted their health care experience. Regarding mental health care, Participant 5 (White, Asian) stated, “My therapist is a queer women of color so I don’t really have to explain much to her, which is exactly why she’s my therapist.” Participant 9 (White) emphasized the importance of HCP gender identity in her experience, noting, “I’ve noticed women, regardless of orientation, are usually a little bit more … empathetic or understanding, so I think … I typically feel safer with a female.”

#### 3.2.5. Social Support

Many participants highlighted the significance of having social support. Participant 19 (Black/African American) described how she sees her friends’ ability to be supportive in comparison to her family:

“When we talk with family, our friends understand because either they have either have been friends with us long enough, they know and have seen some of the things that we encounter um, but I think it’s still really hard for our families to fully conceptualize what [LGBTQ+ health] means.”

#### 3.2.6. Re-Engagement with Care After Bad Experiences

Participants described their resilience in seeking care after having negative experiences in health care. Participant 2 (White) shared her rationale for continuing to seek care:

“It would be stupid to … allow that [other people’s attitudes or behavior] to … stop me from going to the doctor if it’s … really important, and it is because I have all these issues. So yeah, I guess I just remind myself that they’re wrong, and I should still go to the doctor.”

Similarly, Participant 12 (Latina) explained her perspective on re-engaging in care:

“That experience, like my first bad experiences, they weren’t going to be all my experiences. I’m a pretty half-glass-full kind of person, so I was just really hopeful that with time and more research, because … I can call and be like ‘I need a better one.’”

#### 3.2.7. Socio-Economic Advantages

Participants identified socio-economic factors, including consistent health care coverage and other privileges (e.g., transportation), that make it easier for them to engage in health care. Participant 16 (Black/African American) stated, “I’ve been insured my whole life which I recognize is a privilege as I’ve always been able to access most doctors and services.”

Participant 17 (Black/African American) acknowledged the additional factors that aided her ability to access medical care, stating, “Yes, I do have access to those things [that allow for regular health care]. I have the time to take off work, I have a vehicle to get to the provider, and I do have regular health care providers.”

## 4. Discussion

This study aimed to identify barriers and facilitators that SMW have experienced when accessing mental and physical health care through in-depth interviews. Using Andersen and Davidson’s [26] health care access model, contextual and individual barriers and facilitators were identified. A total of six barrier subthemes were identified, including four contextual barrier subthemes and two individual barrier subthemes. Additionally, seven facilitator subthemes were identified, with four contextual facilitator subthemes and three individual facilitator subthemes.

In the sample of 20 SMW, the four contextual barrier subthemes that emerged included discrimination, dominant culture centric (i.e., heterosexual), unsupportive socio-political environment, and lack of patient-centered care. Specific codes demonstrate the unique experience of SMW framed within this model. Aligned with Andersen and Davidson’s [26] health care access model, it was predicted that SMW participants would report discrimination and heterosexism [18] as significant contextual barriers to health care access. Consistent with this hypothesis, discrimination, including LGBTQ+-related, gender-related, race-related, and weight-related discrimination, emerged as a significant barrier subtheme. Though not hypothesized, many participants endorsed unsupportive socio-political barriers, including stigmatizing laws and policies, as well as a lack of LGBTQ+ acceptance in their town. Cultural barriers, including heteronormativity, as well as insufficient HCP knowledge regarding SMW-specific health care needs, also emerged. Consistent with the hypotheses, cultural barriers presented exceptional challenges to obtaining reproductive and mental health care among our sample. Specifically, participants discussed insufficient HCP knowledge regarding SMW’s reproductive and mental health care needs resulting in uniquely invalidating health care encounters. Andresen and Davidson’s [26] model considered provider-related factors as contextual characteristics impacting health care access. Lack of patient-centered care from HCPs emerged as a barrier, including poor bedside manner and HCPs not listening. Specifically, American medical training has only committed to the standards of culturally sensitive and patient-centered practice since 2000, making this area a critical target for continuing education for providers who completed their training before this time [31].

The two individual barrier subthemes that emerged included avoidance or concealment of sexual identity and socio-economic challenges. Socio-economic challenges, including low income and insufficient health insurance coverage, are well captured by Andersen and Davidsen’s [26] health care access model. Andersen and Davidsen [26] point to health beliefs and attitudes as individual factors associated with health care access. The SMW in the current study reported unique identity-specific negative experiences with health care providers. Specifically, heterosexual, culture-centric assumptions held by providers made it more likely that some women in our sample would avoid care. Additionally, while some women in our sample did not report avoiding seeking health care, many reported avoidance of disclosing their sexual identity (i.e., concealment). While identity concealment does not always prevent SMW from seeking care, it may prevent them from receiving validating and patient-centered care, which may be conceptualized as a minority stress experience {16, 17].

The four contextual facilitator subthemes identified were having a supportive socio-political environment, knowing in advance if an HCP is affirming, patient-centered care, and HCPs and patients sharing a similar identity. In terms of contextual facilitators, it was predicted that participants would report positive health care experiences with trusted HCPs and affirmative environments [19,20]. Two subthemes that supported the affirmative environments hypothesis were a supportive socio-political environment, such as having an inclusive health center nearby and a strong LGBTQ+ presence in the participant’s area, and the ability to know in advance whether a health care provider is LGBTQ+-affirming. In addition, patient-centered care, knowing in advance if an HCP is affirming, and shared identity with the HCP are all related to having a positive experience with a trusted HCP. Andresen and Davidson’s [26] model highlights provider-related factors and community characteristics as contextual factors, consistent with the facilitators identified.

The individual facilitator subthemes included social support, re-engagement with health care after negative experiences, and socio-economic advantages. It was predicted that the patient’s resilience would be a facilitator [19,20], which aligns with the re-engagement subtheme that demonstrates patient resilience. Andresen and Davidson [26] also identified a person’s social networks, income, and organization of care (whether or not someone has a regular source of care, the nature of that care, and transportation factors) as individual characteristics in accessing care.

### Strengths, Limitations, and Future Directions

A strength of this study is the use of qualitative research methods with an underrepresented population. This approach is consistent with calls for conducting qualitative research to improve the health of women who are historically underrepresented and underserved [28]. In addition, this study utilized a theoretical framework of health care access [26] to identify unique contextual and individual facilitators and barriers SMW encounter when accessing health care. While facilitators and barriers can be seen as opposites (e.g., consistent vs. inconsistent health insurance coverage), facilitators and barriers can also exist independently of one another. As such, identifying unique facilitators of health care access and experiences is important for developing interventions to address these challenges.

Despite the strengths of this study, there are several limitations that should be noted. First, it is unclear whether the participants attributed all barriers and facilitators to their sexual identity. Heterosexism is directly linked to one’s sexual identity. However, other barriers, such as income, health insurance, and health illiteracy may not be directly linked to sexual identity for all SMW. For example, seven SMW in this study identified low income as a barrier to their health care. Compared to heterosexual women, on average, bisexual women earn less, and lesbian women earn more [32]. Despite income variability within SMW, gay, lesbian, and bisexual women are all more likely to report delaying or not engaging in care due to cost [14]. Although lower incomes among bisexual women may partially explain their delaying care due to cost, it remains unclear what other specific barriers may be driving delayed care due to cost for lesbian women. Additionally, it is unknown whether the women in the current study attributed lower income levels to their sexual identity. Future research may investigate whether SMW perceive specific barriers that are less clearly related to sexual identity (e.g., income, health insurance) as being related to sexual identity in their experience.

Second, 75% of participants selected lesbian as at least one of their sexual identities, with far fewer participants identifying as bisexual and/or pansexual. Therefore, this study may be more representative of the experiences of lesbian women than of other SMW subgroups (i.e., bisexual, pansexual). Future studies could recruit a wider variety of sexual identities and focus on subgroups within the population of SMW. This may be particularly important given that bisexual women are at greater risk for health concerns [12,13] and experience increased challenges when seeking health care [14]. In addition to greater sexual identity representation, future research may aim to recruit individuals with a variety of ability statuses and health complexities. Future studies might also focus on groups with specific diagnoses or health concerns. For example, additional insight may be gained from studying risk and protective factors in SMW with psychological or gynecological health concerns who may routinely experience negative health care experiences.

Third, one of the barriers highlighted by participants was stigmatizing laws and policies. In this study, we did not ask where participants lived. Therefore, we were unable to describe the sample based on the areas in which they live and the laws and policies in those areas. Future research should assess the area in which participants live to better describe the challenges they are facing in an ever-changing socio-political landscape.

## 5. Conclusions

The SMW identified a number of barriers and facilitators to health care, with the subthemes and codes of many barriers rooted in homophobia or heterosexism and the subthemes and codes of many facilitators focused on LGBTQ+ inclusivity. A common barrier to health care for SMW was interfacing with a dominant culture-centric (i.e., heteronormative) health care system characterized by inadequate sexual and gender minority knowledge from HCPs. This highlights the importance of HCPs receiving training on LGBTQ+-affirming practices in general and within their health care specialty. This research underscores the consequences of failing to address LGBTQ+ discrimination in health care settings. It also emphasizes how other forms of discrimination (e.g., gender, race, weight) and the intersection of these identities (e.g., being a Black lesbian woman) impact SMW’s health care experiences. HCPs should utilize patient-centered approaches to reduce stigmatization in their health care practices. Findings suggest that barriers to care may be particularly pronounced in the areas of mental health and sexual and reproductive health care, as disclosure of sexual orientation and utilization of LGBTQ+-affirming health care services may be particularly salient in these health care settings. Finally, this study’s findings highlight the importance of contextual facilitators in accessing health care for this community. Specifically, the efforts made by health care systems and offices to be more inclusive of SMW patients may enhance their access to care. This study helps to increase our understanding of the lived experiences of cisgender SMW, which can help enhance knowledge of health care disparities among SMW and guide interventions for this population.

## Figures and Tables

**Table 1 ijerph-22-00965-t001:** Barriers to health care for SMW.

Subthemes	Codes
Discrimination	LGBTQ+ related discrimination
	Gender-related discrimination
	Weight stigma
	Race-related microaggressions and discrimination
Dominant culture centric	Heteronormativity
	LGBTQ+ health is not something family thinks about
	Inadequate LGBTQ+ knowledge from HCPs
Unsupportive socio-political environment	Town is generally not accepting
	Stigmatizing laws and policies
Lack of patient-centered care	HCPs not listening
	Poor bedside manner
Avoidance/concealment of sexual identity	Only disclosing sexual identity when necessary
	Avoiding treatment or putting off treatment due to previous negative health care experiences
	Avoiding treatment as to not disclose sexual identity
Socio-economic challenges	Low income
	No consistent health insurance coverage

**Table 2 ijerph-22-00965-t002:** Facilitators to health care for SMW.

Subthemes	Codes
Supportive socio-political environment	Inclusive health center nearby (e.g., LGBTQ+, Planned Parenthood, university counseling center)
	Strong LGBTQ+ presence/visibility in local community
Advance identification of LGBTQ-affirming HCPs	Finding HCPs online
	Identification of LGBTQ+ friendly care
	Referrals from LGBTQ+ community
Patient-centered care	HCPs who listen and are not dismissive
	HCPs’ use of inclusive language and questions
	HCPs who initiate conversations about LGBTQ+ topics
HCP identity similar to that of the patient	HCPs who identify as LGBTQ+
	Women-identifying, female-identifying HCPs
	HCPs who identify as people of color
Social support	Supportive friends
	Supportive family
Re-engagement with care after bad experiences (i.e., resilience)	Seeking health care with a different HCP
	Prioritizing health despite prior negative experiences
Socio-economic advantages	Regular HCP and ability to access HCP
	Consistent health insurance coverage

## Data Availability

The datasets presented in this article are not readily available due to protection of participant privacy.

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
