# Peer review of "Barriers and Facilitators to Accessing Mental and Physical Health Care Among Sexual Minority Women: A Qualitative Exploration"

_ijerph, 2025, doi:10.3390/ijerph22060965_

Round 1

Reviewer 1 Report

Comments and Suggestions for Authors

The manuscript “Barriers and Facilitators to Accessing Mental and Physical Health Care among Young Sexual Minority Women: A Qualitative Exploration” focuses on a qualitative study that aimed “to identify the barriers and facilitators that young sexual minority women experience when seeking and accessing mental and physical health care”. The manuscript focuses on an interesting and important topic, and it is quite clear. However, I think that the analysis of the results still needs to be "matured”, and the titles of the themes and subthemes improved. For this reason, I consider that the manuscript should be rejected and resubmitted after major revisions.  I have several suggestions for improving the manuscript that I ask the author to consider.

Introduction - The theoretical introduction is written in a concise and clear way, presenting the social problem under analysis and the objectives of the study as it should be done. Contrary to what I usually do in a qualitative study, at the end of the theoretical Introduction, you also raised hypotheses, but these are well supported by the theoretical framework, which is excellent.

Materials and Methods - First, I suggest that, since the manuscript focuses on a qualitative study, you say "Methodology" instead of "Materials and Methods". Then, I would like to say that this section is a bit confusing, there seems to be information mixed together from the usual various subsections into one. To make it clearer, and so that the study can be replicated, I suggest that the 4 “traditional” subsections be inserted and divided as follows:

2.1. Participants

In this subsection, the participants should be characterized, as is done. But currently, the participants subsection has information from both the Participants subsection and the Procedure subsection, which should not happen.

2.2. Procedure

In this subsection, you should ensure that all information related to the procedure is mentioned, including the approval of your Ethics Committee, that is, the criteria defined to select the participants, how the participants were contacted, whether they were informed of the objectives of the study, where the interviews took place, how long it took for the interview to be carried out (it should be stated at least how long the smaller and larger interviews lasted; or the average), whether the interview was recorded with the consent of the participants, whether the anonymity of the participants was guaranteed, whether the interview was then transcribed verbatim, and so on. The same should be mentioned regarding the brief survey.

2.3. Material (or alternatively Instruments)

This subsection should contain information about all materials or instruments used in the study, as well as all questions or variables under study. For example, where did the interview guide come from, or how was it constructed (e.g., from the theoretical framework?), what questions or variables are being studied, or at least the dimensions covered by the guide used. If you want to insert more detailed information into the text, you can do as you do and refer to "supplementary materials".

2.4. Data Analysis

In this subsection, it should be stated how the interview data will be analysed, as is done. But you need to be more specific. For example, did you use Braun and Clarke's thematic analysis methodology? If yes, what the methodology consists of, what exactly it allows you to do or identify; how the analysis was performed (e.g., did you do a deductive or inductive analysis, or both?); who performed the data analysis, what kind of approach was adopted (e.g., essentialist, constructivist, or other), and so on and so on. Also tell how the survey data was analysed.

In short, if the Methodology section is described in more detail through these 4 subsections, it will become clearer and thus it will be possible for other authors to replicate the study.

Results - Since your manuscript presents the results of a qualitative study, I suggest you say "Findings" instead of "Results."

If you did Braun and Clarke's thematic analysis, I find it hard to believe that so many themes emerged from this analysis, because, generally, the themes are quite broad. For example, “Barriers to Health care for SMW” may be a major theme identified in the interview material, which may then encompass several subthemes.

It seems to me that what you call "Themes" in Table 1 and Table 2 are actually codes and subthemes. For example, "Heteronormativity" or "Not feeling safe" seem like codes to me; "Gender-related discrimination" or "Race-related microaggressions and discrimination" seem to be themes or subthemes of a broader Theme on discrimination.

Furthermore, it also seems that some titles need to be improved, because these should say more than what is done on the theme. For example, I think the title of the second "theme" shown in Table 1 "Problems accessing mental health care" could tell us more about what exactly these Problems in accessing mental health care are. This way we don't know anything, except that there are problems. I hope I have been clear.

In short, the analysis of the results of the corpus of material from the 20 interviews still seems a little "mature".

There are several articles on the internet that can serve as an example of the presentation of results analysed using Braun and Clarke's thematic analysis methodology. See, for example, the following online article from a journal of MDPI:

Lopes, B. G., Marques, A. M., & Santos, M. H. (2024). Gender in Portuguese Firefighters: The Experiences and Strategies of Women. Social Science, 13(8), 431; https://doi.org/10.3390/socsci13080431

Pay particular attention to the presentation of the themes, the presentation of the subthemes and the table with the themes and subthemes.

I suggest that the excerpts from the interviews selected to illustrate each of the themes or subthemes include the sociodemographic information of the participant who mentioned it, and which was included in the analysis of the results (e.g., lesbian woman, white, and so on.)

As the aim of your study is to identify barriers and facilitators that working women experienced when accessing mental and physical health care, I question whether thematic analysis was the best choice of data analysis methodology.

Discussion - If you revise the results section, the discussion will also need to be revised. I believe this section will benefit from this revision, because as it stands, with only three paragraphs, the discussion of the results considering theory is very poor.

In the discussion it is necessary to make it clearer whether the objectives or hypotheses were verified or not.

Author Response

  1. The manuscript “Barriers and Facilitators to Accessing Mental and Physical Health Care among Young Sexual Minority Women: A Qualitative Exploration” focuses on a qualitative study that aimed “to identify the barriers and facilitators that young sexual minority women experience when seeking and accessing mental and physical health care”. The manuscript focuses on an interesting and important topic, and it is quite clear. However, I think that the analysis of the results still needs to be "matured”, and the titles of the themes and subthemes improved. For this reason, I consider that the manuscript should be rejected and resubmitted after major revisions.  I have several suggestions for improving the manuscript that I ask the author to consider.

Response: Thank you for the detailed feedback provided below with suggestions for revision. Below we have responded to each point and have addressed these in the revised manuscript.

  1. Introduction - The theoretical introduction is written in a concise and clear way, presenting the social problem under analysis and the objectives of the study as it should be done. Contrary to what I usually do in a qualitative study, at the end of the theoretical Introduction, you also raised hypotheses, but these are well supported by the theoretical framework, which is excellent.

Response: We are pleased to hear that although not always present in qualitative papers, the hypotheses we developed based on the theoretical framework are well-supported.

  1. Materials and Methods -First, I suggest that, since the manuscript focuses on a qualitative study, you say "Methodology" instead of "Materials and Methods". Then, I would like to say that this section is a bit confusing, there seems to be information mixed together from the usual various subsections into one. To make it clearer, and so that the study can be replicated, I suggest that the 4 “traditional” subsections be inserted and divided as follows:

Response: We appreciate this feedback and the specific details regarding places where information contained in subsections was confusing. We have changed heading 2 to read “Methodology” as suggested by the reviewer (p. 4 line 166). Below, we have responded to each additional point regarding the method section.

  1. 1. Participants. In this subsection, the participants should be characterized, as is done. But currently, the participants subsection has information from both the Participants subsection and the Procedure subsection, which should not happen.

Response: In the revised manuscript, we have removed information regarding recruitment of participants from the participants section and have changed the heading from “participants and recruitment” to only “participants”. This edit can be found on page 4 lines 168-183. We have moved the information about recruiting participants to later in the Procedures section, which begins on page 4 line 185.

  1. 2. Procedure. In this subsection, you should ensure that all information related to the procedure is mentioned, including the approval of your Ethics Committee, that is, the criteria defined to select the participants, how the participants were contacted, whether they were informed of the objectives of the study, where the interviews took place, how long it took for the interview to be carried out (it should be stated at least how long the smaller and larger interviews lasted; or the average), whether the interview was recorded with the consent of the participants, whether the anonymity of the participants was guaranteed, whether the interview was then transcribed verbatim, and so on. The same should be mentioned regarding the brief survey.

Response: Thank you for raising this concern. We added additional details regarding approval from our institutional review board, obtaining participant consent, and procedures regarding interview recordings and transcriptions. These edits can be found on page 4 lines 185-202 of the revised method section.

  1. 3. Material (or alternatively Instruments). This subsection should contain information about all materials or instruments used in the study, as well as all questions or variables under study. For example, where did the interview guide come from, or how was it constructed (e.g., from the theoretical framework?), what questions or variables are being studied, or at least the dimensions covered by the guide used. If you want to insert more detailed information into the text, you can do as you do and refer to "supplementary materials".

Response: We have created a new Materials section where we describe the demographics survey and interview questions briefly. We also include all of these materials in the supplementary materials section and we reference these materials within this section. This new section can be found on page 5 lines 204-210.

  1. 4. Data Analysis. In this subsection, it should be stated how the interview data will be analyzed, as is done. But you need to be more specific. For example, did you use Braun and Clarke's thematic analysis methodology? If yes, what the methodology consists of, what exactly it allows you to do or identify; how the analysis was performed (e.g., did you do a deductive or inductive analysis, or both?); who performed the data analysis, what kind of approach was adopted (e.g., essentialist, constructivist, or other), and so on and so on. Also tell how the survey data was analyzed.

Response: For the analysis of the interviews, we used both inductive and deductive strategies, and we have added this information in the revised manuscript on page 5 lines 213-214. Throughout the Data Analysis section, we have added details to more clearly describe the flow of the data analysis process (page 5 lines 214-220). We have also added that we used SPSS to analyze the demographic survey data, with descriptive statistics being conducted on these data to describe the sample (page 5 lines 221-222).

  1. In short, if the Methodology section is described in more detail through these 4 subsections, it will become clearer and thus it will be possible for other authors to replicate the study.

Response: We appreciate these helpful suggestions and hope that the revised methods section is more clearly described and organized for readers.

  1. Results - Since your manuscript presents the results of a qualitative study, I suggest you say "Findings" instead of "Results."

Response: We have made this requested revision on page 5 line 223. However, if the Editor would prefer that we use more standard language (i.e., Method instead of Methodology, Results instead of Findings) that is consistent with scientific writing, we are happy to change this wording back to our original manuscript.

  1. If you did Braun and Clarke's thematic analysis, I find it hard to believe that so many themes emerged from this analysis, because, generally, the themes are quite broad. For example, “Barriers to Health care for SMW” may be a major theme identified in the interview material, which may then encompass several subthemes. It seems to me that what you call "Themes" in Table 1 and Table 2 are actually codes and subthemes. For example, "Heteronormativity" or "Not feeling safe" seem like codes to me; "Gender-related discrimination" or "Race-related microaggressions and discrimination" seem to be themes or subthemes of a broader Theme on discrimination.

Response: We have revised the labeling of themes, subthemes, and codes throughout the results section of the manuscript and in the tables. In particular, as suggested by the reviewer, in Table 1 we present barriers to healthcare and list subthemes (e.g., discrimination) and codes within this subtheme (e.g., LGBTQ+ related discrimination, gender-related discrimination, weight stigma). We have changed this labeling throughout the results section, which begins on page 5.

  1. Furthermore, it also seems that some titles need to be improved, because these should say more than what is done on the theme. For example, I think the title of the second "theme" shown in Table 1 "Problems accessing mental health care" could tell us more about what exactly these Problems in accessing mental health care are. This way we don't know anything, except that there are problems. I hope I have been clear. In short, the analysis of the results of the corpus of material from the 20 interviews still seems a little "mature".

Response: We appreciate this feedback on clarifying the naming of the subthemes and codes. Our team and have substantially streamlined the naming of subthemes and codes to more appropriately summarize and capture the content of the interviews. These edits were made throughout the results and discussion sections, but most clearly can be seen in Tables 1 and 2. For example, in Table 1 (barriers) instead of conceptualizing the subthemes within “contextual” and “individual” levels, we now have 6 subthemes (discrimination, dominant culture centric, unsupportive socio-political environment, lack of patient-centered care, avoidance/concealment of sexual identity) with 2-4 codes within each of these subthemes (see pages 5-6 for Table 1). We similarly revised Table 2 (facilitators) to now have 7 subthemes with 2-3 codes within each subtheme (see pages 7-8 for Table 2). With these revisions, coupled with the edits to the table format (see Reviewer 1 comment #12 response below), we believe the results are substantially more streamlined and synthesized for readers.

  1. There are several articles on the internet that can serve as an example of the presentation of results analyzed using Braun and Clarke's thematic analysis methodology. See, for example, the following online article from a journal of MDPI: Lopes, B. G., Marques, A. M., & Santos, M. H. (2024). Gender in Portuguese Firefighters: The Experiences and Strategies of Women. Social Science, 13(8), 431; https://doi.org/10.3390/socsci13080431. Pay particular attention to the presentation of the themes, the presentation of the subthemes and the table with the themes and subthemes.

Response: We appreciate this very helpful article suggestion and have modeled our revised Tables 1 and 2 from the Lopes et al (2024) article that was suggested by the reviewer. We are hopeful that this new table format and organization of the results is helpful in conveying the findings from our interviews.

  1. I suggest that the excerpts from the interviews selected to illustrate each of the themes or subthemes include the sociodemographic information of the participant who mentioned it, and which was included in the analysis of the results (e.g., lesbian woman, white, and so on.).

Response: Thank you for the suggestion. While we understand the value of including detailed sociodemographic information with participant quotes, we chose to limit these details to protect confidentiality. We focused on representing diverse perspectives across the dataset and included relevant demographic context within the broader analysis where appropriate.

  1. As the aim of your study is to identify barriers and facilitators that working women experienced when accessing mental and physical health care, I question whether thematic analysis was the best choice of data analysis methodology.

Response: We chose thematic analysis as a way to explore themes that represent the experiences and perceptions of SMW based on the rich responses of interviewees. We also wanted to utilize both inductive and deductive strategies.

  1. Discussion - If you revise the results section, the discussion will also need to be revised. I believe this section will benefit from this revision, because as it stands, with only three paragraphs, the discussion of the results considering theory is very poor.

Response: As noted above, with the substantial revision to the results section, we have also made extensive edits to the discussion section. For example, as mentioned above, in the tables we no longer label contextual and individual factors that come from Andersen and Davidson’s model, but in the discussion we now frame the subthemes within this large model. These revisions can be found, for example, on page 8 lines 427-430, and in other locations throughout the discussion section. These and other edits to the discussion have extended the length to now more fully connect the findings back to both the theoretical framework and hypotheses.

  1. In the discussion it is necessary to make it clearer whether the objectives or hypotheses were verified or not.

Response: This is an excellent point, and we have added information about how the findings from this study support the hypotheses we identified in the introduction and study aims. For example, on page 10 lines 430-432 we now say: “Consistent with this hypothesis, discrimination, including LGBTQ+ related, gender-related, race-related, and weight-related discrimination, emerged as a significant barrier subtheme.” Additional examples of places where we connect findings to the hypotheses can be found on page 10 lines 435-437 and page 11 lines 463-467.

Reviewer 2 Report

Comments and Suggestions for Authors

Thank you for this paper. It was exciting to see a qualitative project on this subject. This is a solid manuscript and I look forward to see it in print.

A few questions and comments:

Pg 3, line 127 - You mention the healthcare access model. Is this the same as the model you introduce in your conceptual framework?

Pg 4, 1.4 Current Study - The shift to 'young women' seems odd here, especially as your sample runs 18-40 with what I interpret to be a mean age of 28.5. Should this read 'young and middle-aged'? Or, does it make sense to qualify the age of the group at all?

Pg 5, 2.3 Data Analysis - It's unclear if you used an inductive or deductive strategy. Maybe both? Please be clear about not just the procedure but the strategy used.

Pg 5, Table 1 - It's difficult to visually determine where new themes begin due to line breaks. Please use a hanging indent or other formatting tool to improve clarity.

Pg 9, 3.2.3 Re-engagement with Care after Bad Experiences - Did you capture any information around disability or health complexity? If so, it would be good to include as a potential moderating factor to resilience/re-engagement with care after bad experiences. 

Pg 9, line 366 - The language here is back to just women, not young women but, on pg 10, line 372, it's young women again. Which is the most accurate way to represent your sample?

Pg 10, 4.1 Strengths, Limitations, and Future Directions - You addressed the major issues with this study - the income effect, bisexual erasure, lack of control for location - clearly and concisely. I almost want something of these in section 1.4 - some version of 'our current study does not directly address income disparities, heavily represents the experience of lesbians, and does not account for variability in policy by location' - so that it doesn't feel like an unanswered question all the way into the discussion section.

Author Response

Reviewer 2:

Thank you for this paper. It was exciting to see a qualitative project on this subject. This is a solid manuscript and I look forward to see it in print.

A few questions and comments:

  1. Pg 3, line 127 - You mention the healthcare access model. Is this the same as the model you introduce in your conceptual framework?

Response: Yes, these are the same. The name and citation were added for clarification.

  1. Pg 4, 1.4 Current Study - The shift to 'young women' seems odd here, especially as your sample runs 18-40 with what I interpret to be a mean age of 28.5. Should this read 'young and middle-aged'? Or, does it make sense to qualify the age of the group at all?

Response: We took out “young” as a descriptor of the sample, so that the age of the group is no longer qualified.

  1. Pg 5, 2.3 Data Analysis - It's unclear if you used an inductive or deductive strategy. Maybe both? Please be clear about not just the procedure but the strategy used.

Response: Yes, both inductive and deductive strategies were used. This has been added to the data analysis section.

  1. 5, Table 1 - It's difficult to visually determine where new themes begin due to line breaks. Please use a hanging indent or other formatting tool to improve clarity.

Response: Individual cells were added to the table for each subtheme and associated codes to improve clarity.

  1. Pg 9, 3.2.3 Re-engagement with Care after Bad Experiences - Did you capture any information around disability or health complexity? If so, it would be good to include as a potential moderating factor to resilience/re-engagement with care after bad experiences. 

Response: Unfortunately, we did not capture information related to disability and health complexity. This has been added to the strengths, limitations, and future directions section as an idea for future research.

  1. Pg 9, line 366 - The language here is back to just women, not young women but, on pg 10, line 372, it's young women again. Which is the most accurate way to represent your sample?

Response: “Young” has been taken out of the description of the sample.

  1. Pg 10, 4.1 Strengths, Limitations, and Future Directions - You addressed the major issues with this study - the income effect, bisexual erasure, lack of control for location - clearly and concisely. I almost want something of these in section 1.4 - some version of 'our current study does not directly address income disparities, heavily represents the experience of lesbians, and does not account for variability in policy by location' - so that it doesn't feel like an unanswered question all the way into the discussion section.

Response: It was not part of the study design to have these limitations. Rather, these limitations emerged after we collected the data. Therefore, we believe it is more appropriate to highlight these limitations in the discussion section.

Round 2

Reviewer 1 Report

Comments and Suggestions for Authors

The manuscript “Barriers and Facilitators to Accessing Mental and Physical Health Care among Young Sexual Minority Women: A Qualitative Exploration” has been revised and is now much better and clearer. For this reason, I think it should be accepted for publication. Before that, I suggest some minor changes.

2.3 Materials - In supplementary materials, I only saw the Demographic Questionnaire, I didn't saw the interview topics. I suggest that you add the interview topics.

2.4 Data Analysis - I still don't understand what methodology was used to analyse the interview material. You said you used thematic analysis, but you didn't say which one exactly. Please clarify, saying which thematic analysis was used, how many phases this analysis has and what each of these phases consists of.

I find it strange that you cannot answer my comment number 13 by providing some sociodemographic information about the participants (e.g., the interview number and the participant's gender), allowing them to remain anonymous. I think this would enrich the analysis.

The manuscript deserves one last careful reading to identify all errors and mistakes that exist throughout the text. For example, the title “3.1.5 Avoidance/concealment” is not complete, or at least not as in Table 1: “Avoidance/concealment of sexual identity”.  Another example: In table 2, you say “Socioeconomic advantages”, then in the text you say “3.2.7 Socio-economic Advantages”. So, a final careful reading would be very good.

One last question: When you say "sexual identity" throughout the text, don't you mean "sexual orientation"?

Author Response

The manuscript “Barriers and Facilitators to Accessing Mental and Physical Health Care among Young Sexual Minority Women: A Qualitative Exploration” has been revised and is now much better and clearer. For this reason, I think it should be accepted for publication. Before that, I suggest some minor changes.

2.3 Materials - In supplementary materials, I only saw the Demographic Questionnaire, I didn't saw the interview topics. I suggest that you add the interview topics.

Response: Thank you for pointing out that the interview topics were missing. They are now included in the supplementary materials along with the demographic questionnaire.

2.4 Data Analysis - I still don't understand what methodology was used to analyse the interview material. You said you used thematic analysis, but you didn't say which one exactly. Please clarify, saying which thematic analysis was used, how many phases this analysis has and what each of these phases consists of.

Response: More information regarding the specific thematic analysis guide used was added. We also added more information about the six phases and clearly identified which phase we were referring to in the text.

I find it strange that you cannot answer my comment number 13 by providing some sociodemographic information about the participants (e.g., the interview number and the participant's gender), allowing them to remain anonymous. I think this would enrich the analysis.

Response: We have included the participant number and the participant’s race/ethnicity. We did not include gender as all of the participants identify as cisgender women.

The manuscript deserves one last careful reading to identify all errors and mistakes that exist throughout the text. For example, the title “3.1.5 Avoidance/concealment” is not complete, or at least not as in Table 1: “Avoidance/concealment of sexual identity”.  Another example: In table 2, you say “Socioeconomic advantages”, then in the text you say “3.2.7 Socio-economic Advantages”. So, a final careful reading would be very good.

Response: Thank you for pointing out these discrepancies. We edited these phrases throughout to text to make them more consistent and did a final reading to search for any other errors.

One last question: When you say "sexual identity" throughout the text, don't you mean "sexual orientation"?

Response: Sexual orientation can refer to one’s attraction, behavior, and identity. The participants in our study were asked to choose sexual identity labels that best applied to them. We did not assess or categorize participants based on elements of sexual attraction or behavior.